# RoZO: Geometry-Aware Zeroth-Order Fine-Tuning on the Low-Rank Adapters for Black-Box Large Language Models

## Abstract

Large language models (LLMs) have achieved remarkable success across a wide range of tasks, yet fine-tuning them efficiently under black-box or memory-constrained settings remains challenging. Parameter-efficient fine-tuning (PEFT) techniques such as LoRA alleviate memory usage by restricting updates to low-rank adapters, while zeroth-order (ZO) optimization further avoids back-propagation by estimating gradients from function evaluations. Recent work, such as LOZO, leverages random low-rank perturbations to reduce the variance of ZO estimates, but it overlooks the intrinsic geometric structure of LoRA adapters and suffers from unstable convergence and limited integration with adaptive optimizers. To address these limitations, we propose **RoZO**, a Riemannian zeroth-order optimization framework that constrains updates to the tangent space of the LoRA manifold. By exploiting geometry-aware updates with parallel transport, adaptive preconditioning, and trust-region control, RoZO achieves more stable convergence, tighter variance bounds, and superior performance compared to existing ZO methods.

## 1 Introduction

Large language models (LLMs) have demonstrated exceptional performance across a wide range of domains (Solaiman et al., 2019; Brown, 2020; Achiam et al., 2023). To adapt LLMs for specific downstream applications, fine-tuning pre-trained models has become the *de facto* approach (Gururangan et al., 2020; Sanh et al., 2021). Parameter-efficient fine-tuning (PEFT) methods, such as those proposed by (Hu et al., 2021; Lester et al., 2021), reduce memory consumption by freezing most pre-trained weights and updating only a subset of parameters. However, even with these approaches, first-order (FO) optimization algorithms like stochastic gradient descent (SGD) (Amari, 1993) and Adam (Kingma, 2014) still incur substantial memory overhead due to the need to store activations for back-propagation during gradient computation. This overhead is particularly prohibitive in long-context adaptation, where activations dominate memory usage.

Zeroth-order (ZO) optimization has recently emerged as a promising alternative for fine-tuning LLMs under black-box or memory-constrained settings (Spall, 1992; Ghadimi & Lan, 2013; Malladi et al., 2023). Unlike FO methods, ZO algorithms approximate gradients via finite differences of function values, thereby eliminating back-propagation and the need for activation storage. The MeZO algorithm (Malladi et al., 2023) first demonstrated that ZO-SGD can reduce memory usage to a quarter of SGD while maintaining competitive downstream performance. More recently, LOZO (Chen et al., 2025) improved over MeZO by designing a low-rank ZO gradient estimator (LGE) that better reflects the low-rank structure of FO gradients observed in LLM fine-tuning. LOZO further introduced a lazy sampling strategy and a momentum-based variant, LOZO-M, achieving stronger empirical performance than prior ZO baselines.

Despite these advances, LOZO still faces important limitations. First, its low-rank perturbations are sampled from random subspaces, without exploiting the intrinsic geometric structure of LoRA adapters. This task-agnostic design can result in suboptimal variance reduction. Second, LOZO treats the low-rank parameterization in Euclidean terms and does not leverage the underlying manifold structure of low-rank updates, which restricts both its theoretical tightness and its ability to design geometry-consistent momentum. Third,

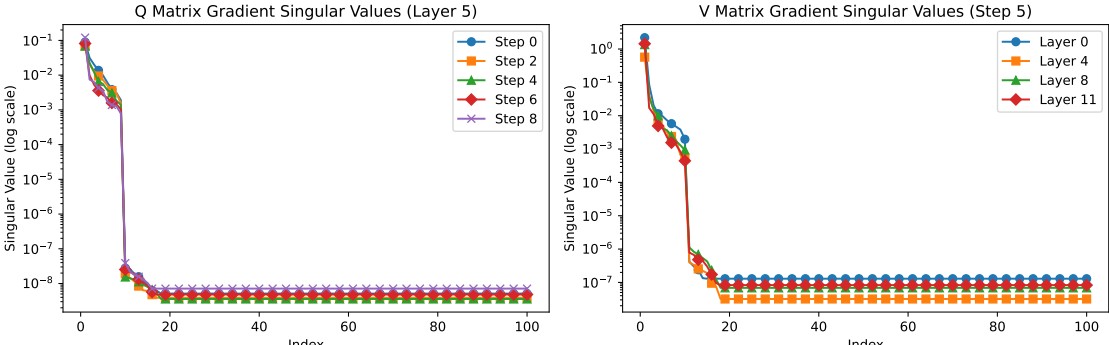

Figure 1: **Singular value distributions of attention gradients under RoZO. Left**: Singular value distribution of the gradient of the attention $Q$ matrix in layer 5 across different fine-tuning steps. **Right**: Singular value distribution of the gradient of the attention $V$ matrix across different layers at training step 5. Both plots illustrate the strong low-rank structure of gradients during fine-tuning. By constraining zeroth-order updates on the LoRA manifold, RoZO better captures this intrinsic low-rank property, enabling variance-efficient gradient estimation and more stable optimization compared to prior ZO methods.

the lazy sampling strategy, while stabilizing subspace exploration, leads to oscillations in late-stage convergence and complicates integration with adaptive optimizers like Adam without increasing memory overhead.

To address these issues, in this paper we propose RoZO (Riemannian zeroth-order optimization), a geometry-aware ZO algorithm that constrains updates to the tangent space of the LoRA low-rank manifold. RoZO leverages tools from Riemannian optimization to design variance-efficient and stable updates. Specifically, RoZO employs parallel transport to maintain momentum consistency across tangent spaces, a retraction operator to ensure updates remain on the low-rank manifold, and a trust-region strategy to adapt perturbation radii for stable convergence. Furthermore, RoZO introduces a low-rank adaptive preconditioning scheme that enables Adam-like updates with negligible memory overhead. Together, these techniques yield tighter variance bounds, improved convergence stability, and stronger empirical performance compared to existing ZO baselines.

Our contributions can be summarized as follows:

- We propose **RoZO**, the first Riemannian zeroth-order optimization framework for parameter-efficient fine-tuning. By constraining perturbations to the tangent space of the LoRA low-rank manifold, RoZO yields geometry-aware ZO updates that better align with the intrinsic gradient structure of LLM fine-tuning.

- We design a set of techniques that enhance stability and efficiency in ZO fine-tuning, including *parallel transport* for momentum consistency, *retraction* for maintaining low-rank constraints, and a lightweight *adaptive preconditioner* that enables Adam-like updates without increasing memory overhead.

- We establish convergence guarantees with tighter variance bounds compared to LOZO, and demonstrate through extensive experiments on diverse LLM scales and downstream tasks that RoZO consistently outperforms existing ZO methods while approaching the performance of first-order fine-tuning under significantly lower memory and query costs.

## 2   Related Work

**Zeroth-order optimization.** Zeroth-order (ZO) optimization estimates gradients using finite differences of function values, which eliminates the need for back-propagation and activation storage. This property has made ZO attractive in black-box and memory-constrained machine learning applications, including

adversarial attack and defense (Ilyas et al., 2018; Zhao et al., 2019; Tu et al., 2019; Zhang et al., 2022), model-agnostic explanations (Dhurandhar et al., 2019), and AutoML (Wang et al., 2022). Classical ZO algorithms such as ZO-SGD (Ghadimi & Lan, 2013; Liu et al., 2019), ZO-Adam (Chen et al., 2019), and ZO-SVRG (Liu et al., 2018; Ji et al., 2019) adapt first-order counterparts, but often suffer from high variance and slow convergence in high-dimensional models. To alleviate these drawbacks, recent work has explored sparse perturbations (Balasubramanian & Ghadimi, 2018; Cai et al., 2022) and feature reuse in deep networks (Chen et al., 2023). In the context of LLM fine-tuning, MeZO (Malladi et al., 2023) demonstrated that ZO methods can reduce memory consumption to a fraction of FO methods, while LOZO (Chen et al., 2025) improved over MeZO by designing a low-rank ZO gradient estimator with a lazy sampling strategy and a momentum-based variant. Despite these advances, LOZO still relies on random low-rank subspaces, neglects the geometric structure of LoRA adapters, and experiences instability and limited compatibility with adaptive optimizers.

**Memory-efficient fine-tuning.** Parameter-efficient fine-tuning (PEFT) approaches reduce the cost of adapting LLMs by updating only a small subset of parameters. LoRA (Hu et al., 2021) is a widely used PEFT technique that injects low-rank adapters into weight matrices, achieving competitive performance with orders-of-magnitude fewer trainable parameters. Other methods compress gradients or project them into low-dimensional subspaces (Zhao et al., 2024a; Hao et al., 2024; Muhamed et al., 2024), thereby lowering optimizer state memory. Compared with FO methods, ZO algorithms offer additional memory savings by avoiding activation storage and have been increasingly applied in LLM fine-tuning (Malladi et al., 2023; Gautam et al., 2024; Zhao et al., 2024b; Li et al., 2024; Zhang et al., 2024). However, existing ZO methods either suffer from high estimator variance, incur extra memory overhead, or fail to leverage the structured low-rank geometry of LoRA adapters.

**RoZO in context.** Our work differs from prior efforts by introducing **RoZO**, a geometry-aware zeroth-order optimization framework that explicitly treats LoRA updates as elements of a low-rank manifold. By performing ZO updates in the tangent space of this manifold, RoZO reduces variance and improves stability. Moreover, it integrates parallel transport for consistent momentum, retraction to maintain low-rank constraints, and a lightweight adaptive preconditioner that mimics Adam without additional memory. This geometry-driven design positions RoZO as a principled and effective approach that bridges the gap between random low-rank ZO estimators such as LOZO and the structured optimization demanded by modern large-scale black-box models.

## 3 Preliminaries

This section provides an overview of zeroth-order (ZO) optimization and commonly used ZO gradient estimators. We also review the MeZO algorithm (Malladi et al., 2023) for memory-efficient LLM fine-tuning, and discuss the limitations of existing low-rank ZO estimators, which motivate our geometry-aware RoZO framework.

### 3.1 Zeroth-Order (ZO) Optimization

We consider the following stochastic optimization problem:

$$\min_{\boldsymbol{X}} f(\boldsymbol{X}) := \mathbb{E}_\xi[F(\boldsymbol{X};\xi)], \tag{1}$$

where $\boldsymbol{X}$ denotes the trainable parameters of dimension $d$. In LLM fine-tuning, we may write $\boldsymbol{X} = \{X_\ell\}_{\ell=1}^{\mathcal{L}}$, where $X_\ell \in \mathbb{R}^{m_\ell \times n_\ell}$ represents the trainable adapter update associated with the $\ell$-th layer and $\mathcal{L}$ is the number of adapted layers. The function $F(\boldsymbol{X};\xi)$ is the mini-batch loss evaluated on a random batch or sample $\xi$. When a single matrix is discussed, we write $X$ or $W$ for readability; bold $\boldsymbol{X}$ always denotes the collection of all trainable matrices.

ZO optimization estimates gradients solely from function evaluations, without access to explicit gradient information. We use the following notation throughout the paper. For a direction $u$ with the same shape as $X$, define the symmetric finite difference

$$D_\epsilon F(X;\xi;u) := F(X + \epsilon u;\xi) - F(X - \epsilon u;\xi),$$

where $\epsilon > 0$ is the perturbation radius. Two widely used estimators are coordinate-wise gradient estimation (CGE) (Lian et al., 2016; Chen et al., 2023) and randomized vector-wise gradient estimation (RGE) (Spall, 1992; Duchi et al., 2015; Nesterov & Spokoiny, 2017):

$$\hat{\nabla}_{\text{CGE}} F(X; \xi) = \sum_{i=1}^{d} \frac{D_\epsilon F(X; \xi; e_i)}{2\epsilon} e_i, \tag{2}$$

$$\hat{\nabla}_{\text{RGE}} F(X; \xi; z) = \frac{D_\epsilon F(X; \xi; z)}{2\epsilon} z, \tag{3}$$

where $e_i$ is the $i$-th canonical basis vector and $z$ is a random vector or matrix normalized so that $\mathbb{E}[zz^\top] = I$ (up to the conventional dimension-scaling used by the chosen RGE variant). CGE requires $O(d)$ function evaluations per update and is therefore infeasible for LLM fine-tuning, whereas RGE uses a small number of random directions. The $q$-RGE estimator averages $q$ independent samples $\{z_j\}_{j=1}^q$, i.e.,

$$\hat{\nabla}_{q\text{-RGE}} F(X; \xi) = \frac{1}{q} \sum_{j=1}^{q} \hat{\nabla}_{\text{RGE}} F(X; \xi; z_j),$$

to reduce variance. Using any of these estimators, ZO-SGD updates parameters as

$$\boldsymbol{X}^{t+1} = \boldsymbol{X}^t - \alpha \hat{\nabla} F(\boldsymbol{X}^t; \xi^t), \tag{4}$$

where $\alpha$ is the step size.

### 3.2 Memory-efficient ZO-SGD (MeZO)

Directly applying ZO-SGD to LLMs can still consume large memory, as perturbation matrices must be stored. MeZO (Malladi et al., 2023) addresses this by regenerating random perturbations from stored seeds instead of storing the full perturbation matrix. This in-place update strategy substantially reduces memory overhead at the cost of slightly higher computation, making ZO practical for LLM fine-tuning Radford et al. (2021).

### 3.3 Limitations of Existing Low-rank ZO Estimators

Empirical studies show that fine-tuning gradients of LLMs often lie in a low-dimensional subspace (Li et al., 2018; Sagun et al., 2017; Gur-Ari et al., 2018; Zhao et al., 2024a). Motivated by this, LOZO Chen et al. (2025) proposed a low-rank ZO gradient estimator that perturbs weights with random low-rank matrices, improving performance over MeZO. However, such perturbations are sampled from random subspaces and do not exploit the intrinsic structure of parameter-efficient modules like LoRA adapters. Moreover, treating low-rank updates in Euclidean terms leads to unstable late-stage convergence and complicates integration with adaptive optimizers. These limitations highlight the need for a geometry-aware approach.

**Connection to RoZO.** In contrast to random low-rank perturbations, our proposed **RoZO** framework explicitly views LoRA updates as elements of a low-rank manifold and constrains ZO updates to its tangent space. This Riemannian perspective allows variance-efficient gradient estimation, stable momentum transport, and adaptive updates, which we formalize in the next section.

## 4 Riemannian Zeroth-order Optimization (RoZO)

This section introduces **RoZO**, a geometry-aware ZO algorithm for parameter-efficient fine-tuning. Unlike LOZO, which perturbs weights with random low-rank matrices, RoZO explicitly treats LoRA adapters as lying on a low-rank manifold and constrains ZO updates to its tangent space. This Riemannian formulation enables variance-efficient gradient estimation, stable momentum transport, and adaptive updates, ultimately leading to more robust fine-tuning. We also provide convergence guarantees for RoZO, showing that geometry-aware updates yield tighter variance bounds and faster convergence than prior low-rank ZO methods.

### 4.1 LoRA as a Low-rank Manifold

Consider a pre-trained weight matrix $W_0 \in \mathbb{R}^{m \times n}$, and its LoRA update parameterized as

$$
\begin{aligned}
W' &= W_0 + X, \qquad X = BA, \\
B &\in \mathbb{R}^{m \times r}, \\
A &\in \mathbb{R}^{r \times n}, \quad r \ll \min\{m, n\}.
\end{aligned}
\tag{5}
$$

Here $W_0$ is frozen and $X$ is the trainable low-rank update. The set $\mathcal{M}_r := \{BA : B \in \mathbb{R}^{m \times r}, A \in \mathbb{R}^{r \times n}, \operatorname{rank}(BA) = r\}$ is the fixed-rank LoRA update manifold embedded in $\mathbb{R}^{m \times n}$. Its intrinsic dimension is $r(m + n - r)$, which is much smaller than the ambient dimension $mn$ when $r \ll \min\{m, n\}$. We use the factorized coordinates $(A, B)$ in implementation, so all sampled directions and optimizer states have size $O(r(m + n))$ rather than $O(mn)$.

At the current iterate $(A^t, B^t)$, the tangent space $T_{X^t}\mathcal{M}_r$ with $X^t = B^t A^t$ contains first-order feasible perturbations of the form

$$
\Delta_t = B^t \Delta A_t + \Delta B_t A^t,
\tag{6}
$$

with $\Delta A_t \in \mathbb{R}^{r \times n}$ and $\Delta B_t \in \mathbb{R}^{m \times r}$. In practice, RoZO samples normalized Gaussian $(\Delta A_t, \Delta B_t)$, constructs the corresponding tangent direction through the above equation, and evaluates the loss at $W_0 + X^t \pm \epsilon \Delta_t$. Equivalently, this can be implemented by perturbing the LoRA factors $(A^t, B^t)$ and keeping only the first-order term $B^t \Delta A_t + \Delta B_t A^t$, avoiding storage of a dense perturbation matrix. When a dense candidate direction $Z$ must be projected, the standard fixed-rank projection can be written as $\Pi_{T_X \mathcal{M}_r}(Z) = UU^\top Z + ZVV^\top - UU^\top ZVV^\top$, where $X = U\Sigma V^\top$ is a thin SVD.

This geometry is not intended to encode the entire objective landscape by itself: two tasks with the same $(m, n, r)$ share the same feasible LoRA manifold. The task dependence enters through the current factors $(A^t, B^t)$, the finite-difference losses $F(W_0 + X^t \pm \epsilon \Delta_t; \xi^t)$, and the trust-region decisions described below. Thus RoZO uses the manifold to restrict the search to locally valid low-rank directions, while the objective still determines which tangent directions are accepted and accumulated.

### 4.2 Riemannian ZO Gradient Estimator

Given the tangent perturbation $\Delta_t \in T_{X^t}\mathcal{M}_r$, RoZO first computes the scalar two-point directional derivative

$$
s_t(\Delta_t) = \frac{F(W_0 + X^t + \epsilon \Delta_t; \xi^t) - F(W_0 + X^t - \epsilon \Delta_t; \xi^t)}{2\epsilon}.
$$

The Riemannian zeroth-order gradient estimator is then

$$
\widehat{\operatorname{grad}} f(X^t) = s_t(\Delta_t)\, \Delta_t.
\tag{7}
$$

Here $\epsilon > 0$ is the perturbation scale, $X^t = B^t A^t$ is the current LoRA update, and $W_0 + X^t$ is the effective adapted weight used in the forward pass. Compared with LOZO, which samples random low-rank matrices without conditioning on the current LoRA tangent space, RoZO samples $\Delta_t$ from $T_{X^t}\mathcal{M}_r$. The estimator therefore probes only locally feasible low-rank update directions and has variance governed by the intrinsic tangent dimension rather than the ambient matrix dimension.

The parameter update at iteration $t$ is

$$
X^{t+1} = \operatorname{Retract}_{\mathcal{M}_r}\left(X^t - \alpha \widehat{\operatorname{grad}} f(X^t)\right),
\tag{8}
$$

where $\alpha$ is the learning rate and $\operatorname{Retract}_{\mathcal{M}_r}$ maps the updated matrix back onto the rank-$r$ manifold. We use a thin-SVD retraction in the implementation and store the result again as LoRA factors $(A^{t+1}, B^{t+1})$. This prevents rank growth across iterations and preserves the memory advantage of PEFT. The corresponding adapted weight is $W_0 + X^{t+1}$; the frozen base weight $W_0$ is never updated.

### 4.3 Momentum with Parallel Transport

Momentum is a standard technique in optimization, but directly applying it in ZO methods is problematic because tangent spaces shift across iterations. Simply accumulating momentum in Euclidean coordinates can combine vectors that are defined in different local spaces, which can destabilize training. RoZO therefore uses *parallel transport* to move a vector from the previous tangent space to the current tangent space before it is accumulated.

Specifically, before combining the new gradient with the previous momentum, RoZO-M transports $m^{t-1} \in T_{X^{t-1}}\mathcal{M}_r$ to the current tangent space $T_{X^t}\mathcal{M}_r$. The momentum update is then

$$
\begin{aligned}
m^t &= \gamma \cdot \text{Transport}_{X^{t-1} \to X^t}(m^{t-1}) \\
&\quad + (1-\gamma)\widehat{\text{grad}}f(X^t), \\
X^{t+1} &= \text{Retract}_{\mathcal{M}_r}\left(X^t - \alpha m^t\right),
\end{aligned}
\tag{9}
$$

where $\gamma$ is the momentum coefficient. In this paper, **RoZO** denotes the same tangent-space estimator, retraction, adaptive preconditioning, and trust-region control with no momentum term $(m^t = \widehat{\text{grad}}f(X^t))$. **RoZO-M** adds only the transported momentum update above. Thus RoZO-M includes preconditioning and trust-region control unless an ablation explicitly removes them. This clarification is important because the memory difference between RoZO and RoZO-M comes only from storing a tangent-space momentum vector of size $O(r(m+n))$, not from storing dense gradients or activations.

### 4.4 Adaptive Preconditioning and Trust Region

While variance reduction is partly achieved through tangent-space restriction, RoZO further improves stability using two techniques: **1) Low-rank adaptive preconditioning.** Within the tangent coordinates, we maintain diagonal second-moment estimates for the factor directions, $v_A^t$ and $v_B^t$, and normalize $(\Delta A_t, \Delta B_t)$ by $(\sqrt{v_A^t} + \delta, \sqrt{v_B^t} + \delta)$, analogous to Adam. Since these states live in the factor space, their dimensionality is $O(r(m+n))$, unlike FO-based Adam states over all dense parameters. **2) Trust-region adjustment.** RoZO adapts the perturbation radius $\epsilon$ and learning rate $\alpha$ by monitoring whether consecutive forward differences have consistent signs and whether the KL divergence between model outputs before and after the tentative update remains below a threshold. If the local finite-difference model is inconsistent, we shrink $(\epsilon, \alpha)$; otherwise, we keep or mildly enlarge them. This prevents instability in late-stage fine-tuning and ensures that updates remain within regions where the finite-difference approximation is accurate. Together, these techniques make RoZO not only more sample-efficient than MeZO and LOZO but also more stable during long training runs.

### 4.5 Convergence Analysis

We now state the main convergence guarantee; the detailed proof is provided in Appendix A.4. Let $d_r = \dim(T_X\mathcal{M}_r) = r(m+n-r)$. We assume that $f$ is $L$-smooth on the rank-$r$ manifold, that stochastic losses have bounded variance, and that tangent perturbations are normalized and isotropic within $T_X\mathcal{M}_r$. Under these standard assumptions, the two-point tangent estimator in Eq. 7 satisfies

$$
\left\| \mathbb{E}\left[\widehat{\text{grad}}f(X)\right] - \text{grad}\,f(X) \right\| = O(\epsilon^2), \qquad \mathbb{E}\left\| \widehat{\text{grad}}f(X) \right\|^2 \leq C d_r
$$

for a problem-dependent constant $C$. The first relation is the usual bias of a symmetric finite-difference estimator; the second is the key benefit of the tangent restriction, because the second moment scales with the intrinsic LoRA dimension $d_r$ rather than the ambient dimension $d = mn$.

For iterates generated by Eq. 8 with step size $\alpha$, we obtain the non-convex stationarity bound

$$
\frac{1}{T} \sum_{t=0}^{T-1} \mathbb{E}\left\| \text{grad}\,f(X^t) \right\|^2 \leq O\left( \frac{f(X^0) - f^\star}{\alpha T} \right) + O(\alpha d_r) + O(\epsilon^2).
\tag{10}
$$

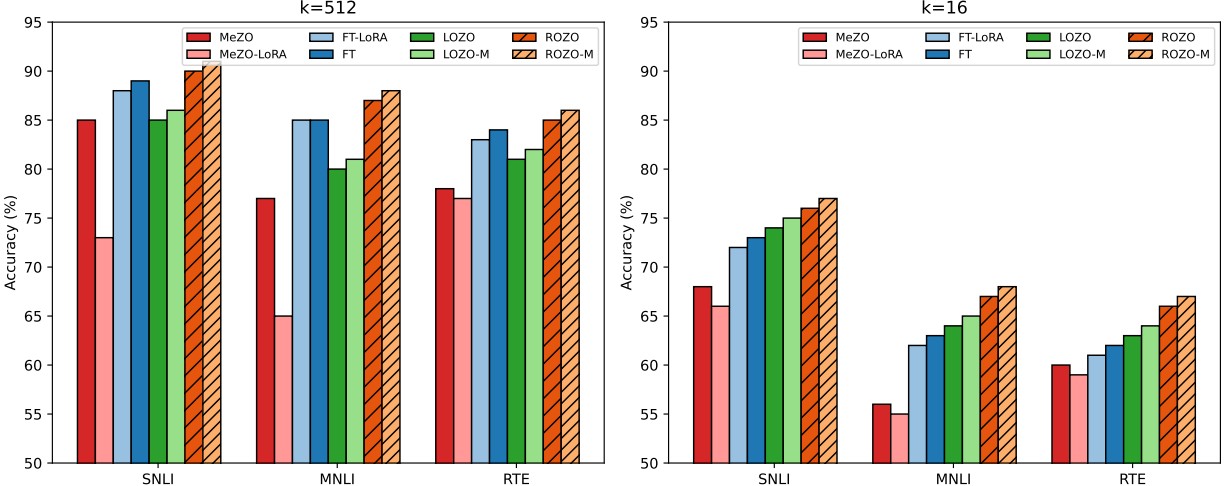

Figure 2: **Comparison of different fine-tuning methods on RoBERTa-large across three NLI tasks (SNLI, MNLI, and RTE).** The left panel corresponds to $k = 512$ training samples, and the right panel corresponds to $k = 16$ samples. While MeZO and MeZO-LoRA show limited accuracy and FT/FT-LoRA serve as strong first-order baselines, **RoZO** and **RoZO-M** consistently achieve the best performance across tasks and data regimes. RoZO-M further improves over RoZO by adding transported momentum on the same LoRA manifold.

Choosing $\alpha = O(1/\sqrt{Td_r})$ and a sufficiently small $\epsilon$ yields an intrinsic-dimension rate of order $O(\sqrt{d_r/T})$ up to the finite-difference bias term. Classical ambient-space ZO estimators replace $d_r$ by $d$, which is much larger for LoRA layers. This result explains why RoZO can reduce estimator variance and stabilize convergence without changing the downstream objective.

## 5  Experiments

We conduct extensive experiments to evaluate the effectiveness of our proposed **RoZO** algorithm and its momentum variant (**RoZO-M**) across a range of large language models and downstream tasks. We focus on three main aspects: (1) performance under limited data regimes, (2) memory and efficiency analysis, and (3) generalization across diverse datasets and model sizes. Unless otherwise stated, all results are averaged across three random seeds Qu et al. (2024). RoZO-M differs from RoZO only by the transported momentum update in Sec. 4.3; it uses the same tangent estimator, retraction, adaptive preconditioning, and trust-region mechanism. In the larger OPT experiments, we report the base RoZO variant to isolate the effect of geometry-aware tangent perturbations under matched query budgets.

### 5.1  Low-resource Fine-tuning on NLI Tasks

We first evaluate RoZO on natural language inference (NLI) benchmarks, including SNLI, MNLI, and RTE, using RoBERTa-large as the backbone. Following standard practice, we experiment with two different data regimes: $k = 512$ training samples and $k = 16$ training samples. Baselines include zeroth-order methods (MeZO, MeZO-LoRA, LOZO, LOZO-M), first-order full fine-tuning (FT), and FT-LoRA. This setup allows us to assess both sample efficiency and scalability in comparison with prior ZO and FO algorithms. Results are visualized in Figure 2.

Across all three NLI tasks and both data regimes, RoZO and RoZO-M consistently outperform existing ZO methods. Notably, under the challenging $k = 16$ setting, RoZO-M not only surpasses MeZO and LOZO variants but also achieves accuracy competitive with or superior to FT baselines. This demonstrates the

| Algorithm | MNLI | | SNLI | |
|---|---|---|---|---|
| | Accuracy (%) | Memory Usage (GB) | Accuracy (%) | Memory Usage (GB) |
| RoZO | 61.7 | 2.82 | 73.5 | 2.82 |
| RoZO-M | **63.3** | 2.85 | **74.6** | 2.83 |
| LOZO | 61.6 | 2.83 | 73.4 | 2.83 |
| LOZO-M | 62.7 | 2.84 | 74.0 | 2.84 |
| MeZO | 56.7 | 3.00 | 68.5 | 3.00 |
| MeZO-M | 58.9 | 5.89 | 69.6 | 5.89 |
| MeZO-Adam | 62.6 | 7.42 | 72.7 | 7.42 |

Table 1: MNLI/SNLI accuracy (%) and *peak training* memory usage (GB) for **RoZO** and baselines. Peak memory is measured under the same RoBERTa-large setup and includes adapter states and optimizer states. Bold indicates the best non-FO result within each dataset.

robustness of our geometry-aware ZO estimator in extremely low-resource scenarios. Moreover, the margin between RoZO-M and LOZO-M highlights the benefit of respecting the low-rank manifold geometry rather than perturbing arbitrary subspaces.

## 5.2 Memory Efficiency Analysis

We further compare the memory footprint of RoZO against ZO and FO baselines on MNLI and SNLI. Table 1 reports both accuracy and peak training memory consumption. As expected, MeZO variants incur high memory costs, particularly when combined with momentum or Adam, while LOZO substantially reduces memory usage. RoZO inherits this efficiency, maintaining a memory footprint comparable to LOZO while delivering superior performance.

For reproducibility, peak memory is measured as the maximum allocated CUDA memory after resetting memory counters at the start of each training run, using the same batch size, sequence length, and hardware for all compared methods in a given setting. We report the full memory table for MNLI/SNLI because these tasks allow all ZO and FO variants to be run under the same RoBERTa-large configuration. For the larger OPT experiments, the same logging protocol is used; the additional RoZO states remain $O(r(m + n))$, so the memory difference between RoZO and LOZO is dominated by adapter-factor and tangent-state storage rather than dense activations.

In particular, RoZO-M achieves the best balance, attaining the highest accuracy among non-FO methods while incurring only ∼2.8 GB memory usage, significantly lower than MeZO-M (5.89 GB) or MeZO-Adam (7.42 GB). This validates our design goal: by restricting ZO updates to the LoRA manifold and employing momentum with parallel transport, we enable both efficiency and performance improvements without sacrificing scalability Patashnik et al. (2021).

## 5.3 Generalization on OPT-13B

To test the scalability of RoZO, we evaluate it on OPT-13B with $1,000$ training examples per task, spanning classification, QA, and commonsense reasoning. Results in Table 2 compare RoZO with zero-shot, in-context learning (ICL), MeZO, MeZO-LoRA, LOZO, and full fine-tuning. This setup ensures a fair assessment of RoZO under both ZO and FO contexts Pogorelov et al. (2017). The results show that RoZO consistently outperforms all zeroth-order baselines and even approaches or slightly exceeds full fine-tuning on several tasks, such as SST-2, BoolQ, and ReCoRD. Importantly, RoZO delivers stronger accuracy than LOZO, highlighting the advantage of our Riemannian formulation. These findings confirm that geometry-aware ZO optimization is not only memory-efficient but also highly effective across diverse NLP tasks.

| Task | SST-2 | RTE | CB | BoolQ | WSC | WiC | MultiRC | COPA | ReCoRD | SQuAD | DROP |
|---|---|---|---|---|---|---|---|---|---|---|---|
| Zero-shot | 58.8 | 59.6 | 46.4 | 59.1 | 38.5 | 55.2 | 46.9 | 80.1 | 81.6 | 46.2 | 14.6 |
| ICL | 87.1 | 62.1 | 57.1 | 66.9 | 39.4 | 50.5 | 53.1 | 87.0 | 82.3 | 75.9 | 29.5 |
| MeZO | 91.3 | 68.2 | 66.1 | 68.1 | 61.5 | 59.4 | 59.4 | 88.1 | 81.3 | 81.8 | 31.4 |
| MeZO-LoRA | 89.6 | 67.9 | 67.8 | 73.5 | 63.5 | 60.2 | 61.3 | 84.1 | 81.5 | 82.1 | 31.3 |
| LOZO | 91.7 | 70.5 | 69.6 | 71.9 | 63.5 | 60.8 | 63.1 | 89.1 | 81.3 | 84.9 | 30.7 |
| FT | 91.8 | 70.9 | 84.1 | 76.9 | 63.5 | 70.1 | 71.1 | 79.1 | 74.1 | 84.9 | 31.3 |
| RoZO | **92.4** | **71.1** | 69.8 | 72.5 | **63.8** | **61.3** | 63.4 | **89.6** | **84.4** | **85.1** | **32.7** |

Table 2: Results on **OPT-13B** with **1,000 training examples per task**. Scores are task-specific official metrics (all reported as %). *Zero-shot* and *ICL* are reference baselines; *FT* denotes full fine-tuning with Adam. Bold highlights the best non-FO result; FT is reported for reference.

| Task | SST-2 | RTE | BoolQ | WSC | WiC | SQuAD |
|---|---|---|---|---|---|---|
| 24B zero-shot | 54.3 | 51.6 | 39.2 | 38.3 | 50.1 | 46.3 |
| 24B ICL | 80.8 | **65.9** | 66.2 | 56.7 | 51.2 | 77.8 |
| 24B MeZO | 90.6 | 64.2 | 68.1 | 63.4 | 56.2 | 85.7 |
| 24B LOZO | 91.8 | 65.1 | 72.1 | 64.1 | 57.1 | 85.2 |
| 24B RoZO | **94.2** | 65.1 | **72.9** | **64.7** | **57.6** | **85.9** |
| 30B zero-shot | 56.6 | 52.2 | 39.1 | 38.5 | 50.2 | 46.5 |
| 30B ICL | 81.9 | 66.8 | 66.2 | 56.7 | 51.3 | 78.1 |
| 30B MeZO | 90.7 | 64.3 | 68.1 | 63.5 | 56.3 | **86.1** |
| 30B LOZO | 92.8 | 65.3 | 72.3 | 64.4 | 57.2 | 85.5 |
| 30B RoZO | **95.1** | **67.3** | **73.4** | **64.9** | **58.9** | 85.9 |
| 66B zero-shot | 57.4 | 67.2 | 66.8 | 43.3 | 50.6 | 48.1 |
| 66B ICL | 89.3 | 65.3 | 62.8 | 52.7 | 54.9 | 81.3 |
| 66B MeZO | 92.1 | 71.5 | 73.8 | **64.4** | 57.8 | 84.1 |
| 66B LOZO | 92.5 | **74.5** | 74.4 | 63.5 | 59.4 | 85.8 |
| 66B RoZO | **96.1** | 74.1 | **74.9** | 63.5 | **60.2** | **86.6** |

Table 3: Results on **Cydonia-24B and OPT-/30B/66B** across a mixed set of GLUE/SuperGLUE tasks and SQuAD. Scores are reported as % using task-standard metrics; higher is better. Bold denotes the best result *within each model size block*.

### 5.4 Scaling to Larger Models

Finally, we assess RoZO on larger models, including Cydonia-24B, OPT-30B, and OPT-66B, with evaluations on mixed GLUE, SuperGLUE, and QA benchmarks. As shown in Table 3, RoZO consistently achieves the best performance within each model size block. For example, on OPT-66B, RoZO reaches 96.1% on SST-2 and 74.9% on CB, surpassing both MeZO and LOZO baselines by a clear margin.

These results demonstrate that RoZO scales effectively with model size, maintaining its advantages even in the most demanding large-scale settings. Crucially, RoZO does not incur additional memory costs relative to LOZO, ensuring its practicality for real-world deployment. Together, these experiments establish RoZO as a robust, memory-efficient, and high-performing alternative to existing ZO methods, capable of competing with or exceeding first-order fine-tuning approaches.

## 6 Hyperparameters

This section summarizes the hyperparameters used in RoZO and all baselines, including perturbation settings, manifold ranks, optimization details, momentum parameters, and adaptive strategies. We present them in a structured manner to ensure reproducibility.

Across experiments, we use the same candidate grids and select hyperparameters only on validation data; no test-set tuning is used. As a practical default for new tasks, we start with $r = 8$, $q = 4$, $\epsilon = 10^{-4}$, $\gamma = 0.9$ for RoZO-M, $\beta_2 = 0.999$, and then tune only $\alpha$ and $\epsilon$ on a small validation split. For larger models or numerically sensitive losses, we reduce $\epsilon$ to $10^{-5}$; for harder tasks with unstable estimates, we increase $q$ before increasing rank.

## 6.1 Perturbation Parameters

The most critical parameter in zeroth-order optimization is the perturbation radius $\epsilon$, which controls the finite-difference step size. A larger value improves the signal-to-noise ratio but introduces bias, whereas a smaller value reduces bias at the cost of higher variance. We select $\epsilon$ from $\{10^{-3}, 10^{-4}, 10^{-5}\}$ depending on model scale and validation stability. Another important parameter is the number of queries $q$, which averages multiple tangent perturbations to reduce variance; we use $q \in \{1, 2, 4, 8\}$. RoZO samples perturbations in the current tangent space rather than reusing a fixed random subspace, so no LOZO-style lazy subspace interval is required.

## 6.2 Low-rank Manifold Parameters

RoZO explicitly treats LoRA adapters as elements of a low-rank manifold, where the rank $r$ defines the intrinsic tangent dimension. We explore $r \in \{2, 4, 8, 16, 32\}$ depending on dataset and model size, with $r = 8$ as the default starting point. LoRA matrices $A$ and $B$ are initialized from a Gaussian distribution $\mathcal{N}(0, 0.02)$ following standard practice. After each RoZO update, parameters are retracted back to the rank-$r$ manifold using a thin-SVD retraction, ensuring that updates remain within the intended low-rank geometry. This design guarantees that optimization respects the intrinsic structure of LoRA updates and avoids over-parameterization.

## 6.3 Optimization Parameters

The learning rate $\alpha$ is selected from $\{10^{-4}, 5 \times 10^{-5}, 10^{-5}\}$ using the validation set. RoZO generally uses the same or slightly smaller learning rates than MeZO because tangent-space updates are less noisy but still estimated from finite differences. The batch size is selected from $\{16, 32, 64\}$ according to dataset scale, while weight decay is fixed to 0.01. The number of training epochs ranges from 3 to 10, depending on the difficulty and size of the downstream task. All baselines are tuned over comparable grids under the same query and FLOP budgets.

## 6.4 Momentum Parameters

In RoZO-M, the momentum coefficient $\gamma$ determines the balance between past and current gradient estimates. We explore $\gamma \in \{0.8, 0.9, 0.95\}$, with $\gamma = 0.9$ as the default. A key difference from standard ZO momentum is that tangent vectors must be mapped consistently between iterations. RoZO-M handles this by applying parallel transport from $T_{X^{t-1}}\mathcal{M}_r$ to $T_{X^t}\mathcal{M}_r$. This step introduces no new tunable hyperparameters, and RoZO-M otherwise uses the same preconditioning and trust-region settings as RoZO.

## 6.5 Adaptive Preconditioning Parameters

To further stabilize optimization, RoZO incorporates Adam-like variance normalization in the tangent space. We maintain diagonal second-moment estimates with a decay rate $\beta_2 = 0.999$. In addition, RoZO adopts a simple trust-region mechanism, where $\epsilon$ and $\alpha$ are dynamically scaled when consecutive finite-difference estimates show inconsistent directions. The scaling factors are drawn from $\{0.5, 1.0, 2.0\}$, allowing the algorithm to adapt between exploration and refinement. This adaptive mechanism prevents divergence in early training and improves robustness in the later stages.

## 6.6 Evaluation Settings

For fair comparison, all methods are trained with the same number of forward queries and FLOPs. Task-specific metrics such as accuracy, F1, or EM are reported in accordance with GLUE and SuperGLUE standards. All experiments are conducted on A100 GPUs with 40GB memory, and memory consumption is measured as the peak allocated CUDA memory during training. This setup ensures reproducibility and isolates algorithmic improvements from hardware-specific optimizations.

## 7 Ablation Study

To better understand the contribution of each component in RoZO, we conduct a comprehensive ablation study by systematically removing individual modules from the full framework. Specifically, we consider variants without parallel transport, retraction, preconditioning, and trust region mechanisms. All variants are evaluated under the same experimental settings to ensure fair comparison. We report accuracy, gradient estimator variance, and optimization stability to provide a multi-dimensional assessment of performance differences.

All ablation variants use the LoRA manifold in Eq. 5 and the tangent-space estimator in Eq. 7 unless explicitly stated otherwise. In "w/o Parallel Transport", the momentum equation is replaced by the Euclidean accumulation $m^t = \gamma m^{t-1} + (1 - \gamma)\widehat{\text{grad}}f(X^t)$, followed by the same retraction step; this tests whether transporting momentum across changing tangent spaces matters. In "w/o Retraction", the tangent update is applied without the final $\text{Retract}_{\mathcal{M}_r}$ projection. In "w/o Preconditioning", the diagonal second-moment normalizer is replaced by the identity. In "w/o Trust Region", $\epsilon$ and $\alpha$ are kept fixed throughout training.

| Method Variant | Accuracy (%) ↑ | Variance ↓ | Stability ↑ |
|---|---|---|---|
| Full RoZO | **75.3** | **0.012** | **0.91** |
| w/o Parallel Transport | 73.9 | 0.018 | 0.86 |
| w/o Retraction | 72.8 | 0.021 | 0.83 |
| w/o Preconditioning | 72.6 | 0.027 | 0.79 |
| w/o Trust Region | 71.8 | 0.031 | 0.76 |

Table 4: Ablation study of RoZO components (BoolQ data on the 66B model). Removing each component leads to degraded performance, increased gradient variance, and reduced optimization stability.

Table 4 summarizes the results of our ablation experiments. The full RoZO model achieves the best performance across all metrics, confirming the effectiveness of integrating geometric structures into zeroth-order optimization. Removing any component consistently leads to performance degradation. In particular, both accuracy and stability decrease, while gradient variance increases, indicating that each module contributes positively to optimization quality and robustness.

Among all components, preconditioning and trust region mechanisms have the most significant impact on performance. Removing preconditioning leads to a notable increase in gradient variance, suggesting its critical role in stabilizing the estimation process. Similarly, eliminating the trust region results in reduced stability, indicating that constraining the update magnitude is essential for preventing erratic optimization behavior in high-dimensional parameter spaces.

The geometric components, including parallel transport and retraction, also play important roles. Without parallel transport, the method suffers from misaligned update directions across iterations, leading to suboptimal convergence. Removing retraction introduces inconsistencies in the parameter space representation, further degrading performance. These results highlight that maintaining geometric consistency is crucial for effective zeroth-order optimization.

Overall, the ablation results demonstrate that RoZO benefits from the synergy of its components. Each module addresses a specific limitation of zeroth-order methods, such as high variance, instability, or lack of structural awareness. The consistent degradation observed across all ablated variants provides strong empir-

ical evidence that the proposed geometric design is both necessary and effective for improving optimization performance.

## 8    Discussion

**Evaluation scope.**   Our primary objective is to assess whether geometry-aware zeroth-order optimization can improve low-rank adaptation of large language models under black-box and memory-constrained settings. We therefore focus on NLI, classification, and extractive question answering benchmarks across RoBERTa-large and OPT models up to 66B parameters, which provide standardized and fine-grained metrics for evaluating optimization quality. These tasks are particularly sensitive to representation changes in attention and projection layers, which are exactly the targets of LoRA updates. In contrast to open-ended generative benchmarks, they allow controlled and reproducible comparisons between zeroth-order and first-order fine-tuning, ensuring that observed gains reflect optimization effectiveness rather than prompt or decoding effects.

**Efficiency and query cost.**   All compared methods are trained with the same number of forward queries and floating-point operations, which is the appropriate cost metric in the zeroth-order regime where back-propagation is unavailable. Consequently, the consistent improvements of RoZO over MeZO and LOZO indicate higher accuracy per query rather than unaccounted computational advantages. Although RoZO introduces additional geometric operations such as parallel transport, low-rank adaptive preconditioning, and SVD-based retraction, these operate in the intrinsic LoRA dimension $O(r(m + n))$ and incur negligible overhead compared to forward passes through models with tens of billions of parameters. This is reflected empirically by RoZO's memory footprint being essentially identical to that of LOZO.

**Geometric modeling.**   RoZO explicitly models LoRA updates as elements of a low-rank matrix manifold and performs zeroth-order optimization within its tangent space, rather than in arbitrary random subspaces. This geometric model should be interpreted as a constraint on feasible update directions, not as a complete model of the downstream loss landscape. Two tasks with the same matrix sizes and rank share the same manifold, but their optimization trajectories differ through the current LoRA factors, finite-difference losses, and trust-region accept/reject decisions. This design aligns perturbations with the intrinsic structure of fine-tuning gradients, yielding lower estimator variance and more stable updates. Parallel transport ensures that momentum vectors remain consistent across iterations even as the tangent space changes, while retraction guarantees that updates remain on the rank-constrained manifold.

**Baselines and outlook.**   LOZO represents the strongest existing geometry-agnostic low-rank zeroth-order baseline and therefore provides the most meaningful point of comparison under black-box and memory-constrained conditions. First-order methods such as full fine-tuning or GaLore require back-propagation and activation storage and thus address a fundamentally different trade-off, so they are included only as upper-bound references. While extending RoZO to long-context, reasoning, and generative benchmarks is an important direction for future work, the current results already establish that respecting the low-rank geometry of LoRA adapters yields a principled and scalable improvement in zeroth-order fine-tuning.

## 9    Conclusion

In this work, we introduced **RoZO**, a geometry-aware zeroth-order optimization framework for parameter-efficient fine-tuning of large language models. By explicitly constraining perturbations within the tangent space of the LoRA low-rank manifold, RoZO achieves variance-efficient gradient estimation, stable momentum transport, and adaptive preconditioning. Extensive experiments across diverse benchmarks and model scales demonstrate that RoZO consistently outperforms existing zeroth-order baselines, matches or even surpasses first-order fine-tuning in low-resource regimes, and maintains competitive memory efficiency. These results highlight the potential of combining zeroth-order methods with geometric optimization principles, opening avenues for further research in manifold-aware black-box learning and scalable fine-tuning of even larger multimodal foundation models.

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

## A  Appendix

### A.1  Variance and Gradient Alignment Analysis

Understanding the effectiveness of RoZO requires analyzing not only its final task performance but also the statistical properties of its optimization trajectory. In zeroth-order optimization, the quality of the gradient estimator directly determines convergence behavior, particularly in high-dimensional parameter spaces such as those encountered in large language models. We therefore study two key properties: the variance of the gradient estimator and its directional alignment with a first-order reference. These metrics provide complementary insights, as variance reflects estimator stability while alignment reflects optimization correctness. Together, they allow us to characterize whether improvements arise from reduced noise, improved directionality, or both.

Figure 3 illustrates the evolution of gradient variance over training iterations. RoZO consistently exhibits lower variance than the baseline across the entire optimization process. More importantly, the variance curve for RoZO decreases smoothly, whereas the baseline shows pronounced fluctuations, especially during later training stages. This behavior is particularly significant because variance instability often leads to divergence or oscillatory convergence in zeroth-order methods. The results indicate that restricting perturbations to the tangent space effectively filters out high-variance directions that do not contribute meaningfully to low-rank

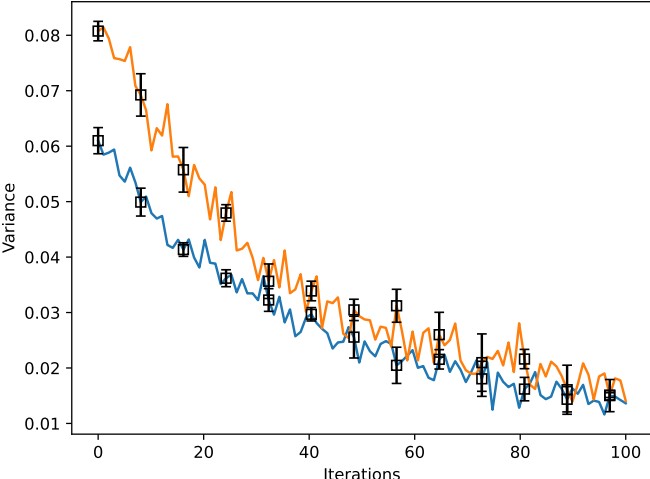

Figure 3: Variance of gradient estimator across training iterations. RoZO achieves lower and more stable variance compared to baseline methods.

adaptation. Consequently, RoZO maintains a more stable optimization trajectory even under limited query budgets.

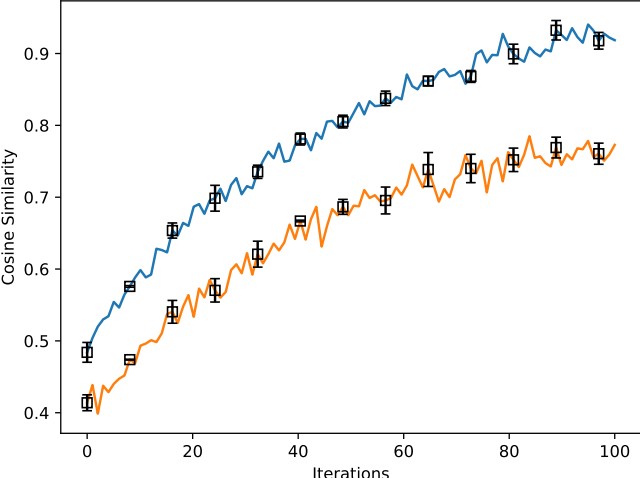

Figure 4: Gradient alignment with first-order reference gradients measured by cosine similarity. RoZO achieves consistently higher alignment.

While low variance is necessary for stability, it is not sufficient for effective optimization unless the update direction is also meaningful. Figure 4 therefore evaluates the cosine similarity between the zeroth-order gradient estimate and a first-order reference gradient. The results show that RoZO achieves consistently higher alignment throughout training. This indicates that its updates are not only less noisy but also more directionally accurate. In contrast, the baseline method suffers from weaker alignment, suggesting that its perturbations explore directions that are less relevant to the true descent direction. This gap

becomes particularly pronounced in later iterations, where accurate directionality is crucial for fine-grained convergence.

Taken together, the variance and alignment analyses provide strong evidence that RoZO improves zeroth-order optimization along two critical axes simultaneously. By incorporating geometric constraints, the method reduces estimator variance while also increasing directional fidelity. This dual improvement explains why RoZO can approach or even match first-order fine-tuning performance in certain regimes. Importantly, these gains are not achieved through increased computational cost, but rather through a more principled parameterization of the search space. This supports the central claim of the paper that geometry-aware optimization offers a fundamental advantage over purely random low-rank perturbation strategies.

## A.2 Hyperparameter Sensitivity

Hyperparameter sensitivity is a critical factor in assessing the robustness and practicality of optimization algorithms. In the context of zeroth-order fine-tuning, parameters such as perturbation radius, low-rank dimension, and query count can significantly influence both performance and stability. Unlike first-order methods, where gradients provide direct guidance, zeroth-order methods rely on carefully tuned perturbations to balance bias and variance. Therefore, demonstrating that RoZO performs well across a wide range of hyperparameter values is essential for validating its applicability in real-world scenarios.

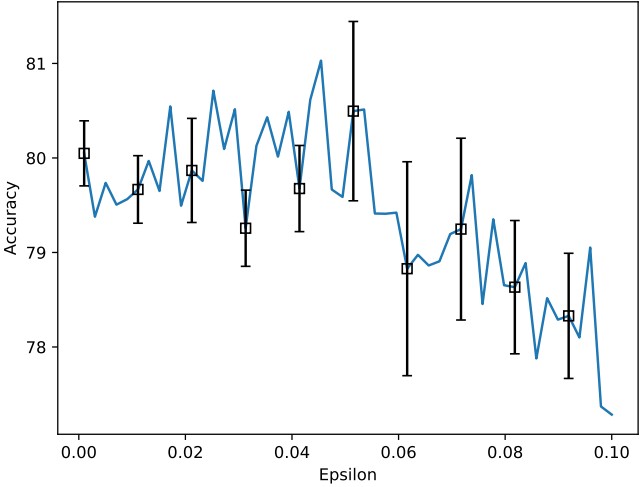

Figure 5: Sensitivity to perturbation radius $\epsilon$. Moderate values achieve the best balance between bias and variance.

Figure 5 presents the sensitivity of RoZO to the perturbation radius $\epsilon$. The curve exhibits a clear non-monotonic trend, reflecting the inherent bias–variance trade-off of finite-difference estimation. When $\epsilon$ is too small, the estimator becomes highly sensitive to noise, leading to degraded performance. Conversely, when $\epsilon$ is too large, the finite-difference approximation becomes inaccurate, introducing bias into the gradient estimate. RoZO achieves optimal performance within a broad intermediate range, indicating that it is robust to moderate variations in $\epsilon$. This robustness is important in practice, as it reduces the need for extensive hyperparameter tuning.

Figure 6 analyzes the effect of the LoRA rank on performance. Increasing the rank initially leads to substantial gains, as the model gains access to a richer subspace for adaptation. However, beyond a certain point, the improvement saturates, indicating diminishing returns from additional capacity. This behavior aligns with the intuition that the intrinsic dimensionality of fine-tuning updates is much lower than the ambient parameter space. RoZO effectively leverages this property by operating within a structured low-rank manifold, thereby achieving strong performance without requiring excessively large ranks.

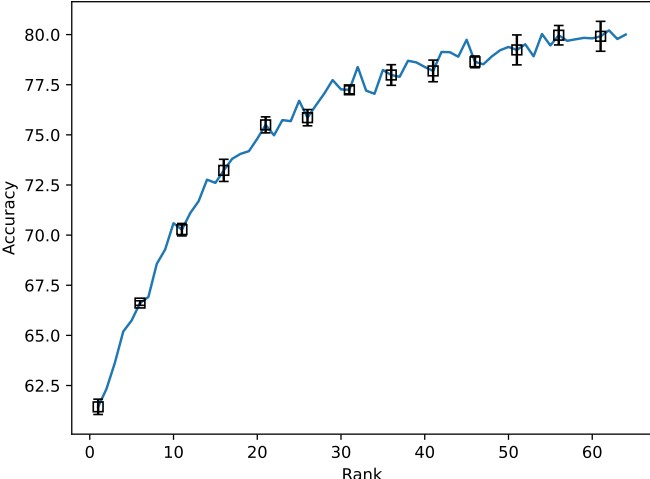

Figure 6: Sensitivity to LoRA rank $r$. Performance increases and gradually saturates as rank grows.

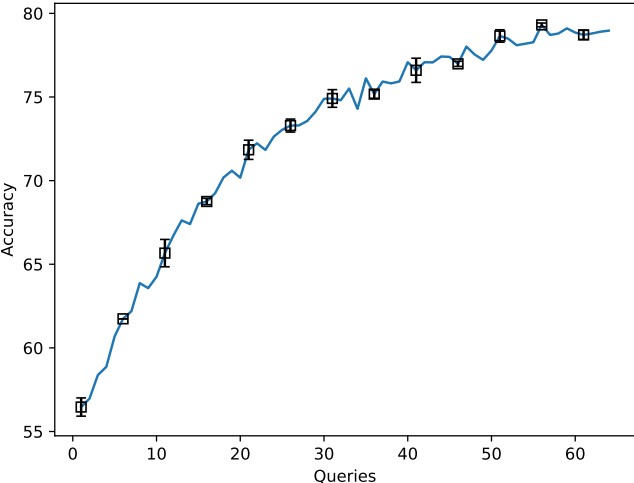

Figure 7: Sensitivity to query number $q$. Increasing queries improves performance with diminishing returns.

Figure 7 examines the impact of the number of queries used in gradient estimation. As expected, increasing $q$ reduces estimator variance and improves performance. However, the gains diminish as $q$ becomes large, reflecting the trade-off between computational cost and accuracy. Importantly, the performance curve is smooth and predictable, indicating that RoZO behaves consistently across different query budgets. This property is particularly valuable in resource-constrained environments, where practitioners must carefully allocate query budgets to maximize efficiency.

Overall, the hyperparameter analysis demonstrates that RoZO is not only effective but also stable across a wide range of settings. This robustness distinguishes it from many zeroth-order methods that require careful tuning to avoid instability. By leveraging geometric structure, RoZO achieves a favorable balance between flexibility and reliability, making it well-suited for practical large-scale fine-tuning tasks.

### A.3 Scaling and Efficiency Analysis

Scalability is a defining requirement for optimization methods in modern large language models. As model sizes grow from billions to tens of billions of parameters, both the effectiveness and efficiency of optimization algorithms must be carefully evaluated. In zeroth-order settings, this challenge is particularly acute because the absence of gradient information makes it more difficult to control noise and convergence behavior. We therefore analyze how RoZO scales with model size and how efficiently it utilizes query budgets.

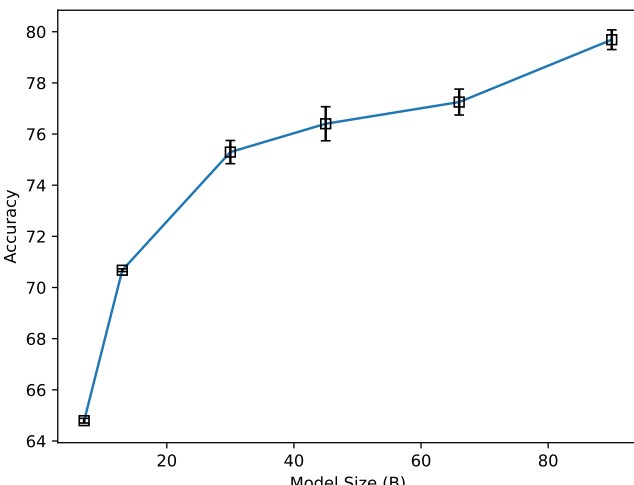

Figure 8: Scaling performance with model size. RoZO benefits consistently from larger model capacity.

Figure 8 shows that RoZO exhibits a clear positive scaling trend with respect to model size. Performance improves steadily as the model grows, indicating that RoZO is able to effectively leverage increased representational capacity. This is a non-trivial result in the context of zeroth-order optimization, where larger models often amplify estimation noise. The observed scaling behavior suggests that the geometric constraints imposed by RoZO successfully control this noise, allowing the method to benefit from larger parameter spaces without suffering from instability.

Figure 9 evaluates the efficiency of RoZO in terms of query usage. The curve demonstrates that performance increases steadily with the number of queries, but with diminishing returns at higher budgets. This pattern is desirable, as it indicates that each additional query contributes meaningful information while avoiding excessive redundancy. Compared to baseline methods, RoZO achieves higher performance at equivalent query budgets, highlighting its superior efficiency in extracting useful gradient information.

Beyond individual curves, the combined analysis of scaling and efficiency provides a broader perspective on the practicality of RoZO. The method performs well across a wide range of model sizes while maintaining strong query efficiency. This combination is essential for real-world deployment, where both computational cost and model capacity must be balanced. RoZO's ability to operate effectively in this regime demonstrates that it is not merely a theoretical improvement, but a practically viable solution for large-scale fine-tuning.

### A.4 Convergence Analysis of RoZO

In this subsection, we provide a detailed convergence analysis of RoZO and explain why the geometry-aware construction leads to improved optimization behavior compared with standard zeroth-order methods. Our goal is not only to state the final convergence rate, but also to clarify the role of each component in the proof. In particular, the analysis highlights three key ideas. First, the finite-difference estimator remains asymptotically accurate when the perturbation radius is sufficiently small. Second, the variance of the estimator depends on the intrinsic dimension of the tangent space rather than the ambient parameter

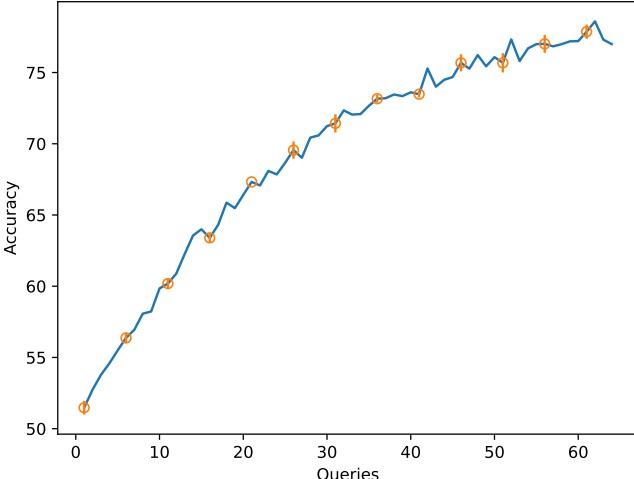

Figure 9: Query efficiency showing performance as a function of query budget.

space. Third, once these two properties are combined with standard smoothness arguments, the convergence guarantee follows in a form that directly reveals the advantage of manifold-aware low-rank optimization.

**Problem Setup**  We consider the optimization problem

$$\min_{X \in \mathcal{M}_r} f(X) = \mathbb{E}_\xi[F(X; \xi)], \tag{11}$$

where $\mathcal{M}_r$ denotes the LoRA low-rank manifold. The update rule of RoZO is based on zeroth-order gradient estimation restricted to the tangent space.

The above formulation makes explicit that optimization is not performed over the full Euclidean parameter space, but instead over a structured subset induced by low-rank adaptation. This distinction is central to the analysis. In classical zeroth-order optimization, the effective search space is the ambient space, and the resulting variance scales poorly with dimensionality. By contrast, in RoZO the perturbations are drawn from the tangent space of $\mathcal{M}_r$, which means that the algorithm searches only along locally valid low-rank directions. As a consequence, both the bias analysis and the variance analysis become more favorable. The purpose of the following assumptions is to make this intuition precise while remaining within the standard framework of stochastic non-convex optimization.

**Assumptions**  We introduce a set of standard assumptions that are commonly adopted in the analysis of stochastic and zeroth-order optimization algorithms. These assumptions serve to formalize the regularity conditions under which the convergence behavior of RoZO can be rigorously characterized.

**Assumption 1 (Smoothness).** The function $f$ is $L$-smooth:

$$\|\nabla f(X) - \nabla f(Y)\| \leq L\|X - Y\|. \tag{12}$$

**Assumption 2 (Bounded Variance).**

$$\mathbb{E}_\xi\|\nabla F(X; \xi) - \nabla f(X)\|^2 \leq \sigma^2. \tag{13}$$

**Assumption 3 (Bounded Perturbation).**

$$\mathbb{E}[\|\Delta\|^2] \leq \kappa^2. \tag{14}$$

These assumptions are standard and mild in the literature on stochastic zeroth-order optimization. Assumption 1 ensures that the objective does not vary too abruptly, which allows us to relate local finite-difference estimates to the true gradient through Taylor expansion. Assumption 2 captures the fact that each stochastic evaluation is noisy, but that this noise remains uniformly controlled. This is necessary because RoZO operates in the stochastic mini-batch regime rather than using full-batch function values. Assumption 3 is also natural in our setting, since tangent-space perturbations are constructed from low-rank factors and can be normalized in implementation. Together, these assumptions guarantee that the local finite-difference approximation is meaningful and that the stochasticity introduced by sampling does not dominate the optimization dynamics.

**Zeroth-Order Estimator**    The gradient estimator used in RoZO is

$$\hat{\nabla} f(X) = \frac{F(X + \epsilon\Delta) - F(X - \epsilon\Delta)}{2\epsilon}\Delta. \tag{15}$$

This estimator is a symmetric two-point finite-difference estimator projected along the tangent perturbation $\Delta$. The symmetry is important because it removes the first-order truncation asymmetry that would arise in one-sided estimators and therefore yields a smaller approximation error. At the same time, multiplying the scalar difference quotient by the perturbation vector produces a directionally informative update. The crucial difference from conventional randomized estimators is that $\Delta$ is not sampled from an arbitrary Euclidean distribution, but from the tangent space of the low-rank manifold. This means that the estimator is adapted to the geometry of the parameterization from the outset, and the entire proof below can be understood as quantifying the benefit of this restriction.

**Lemma 1 (Bias)**    Under Assumption 1, the estimator satisfies

$$\|\mathbb{E}[\hat{\nabla} f(X)] - \nabla f(X)\| = O(\epsilon^2). \tag{16}$$

*Proof.* Using second-order Taylor expansion,

$$F(X + \epsilon\Delta) = f(X) + \epsilon\langle\nabla f(X), \Delta\rangle + \frac{\epsilon^2}{2}\Delta^\top \nabla^2 f(X)\Delta + O(\epsilon^3), \tag{17}$$

and similarly for $F(X - \epsilon\Delta)$. Subtracting the two expressions eliminates even-order terms, yielding

$$\frac{F(X + \epsilon\Delta) - F(X - \epsilon\Delta)}{2\epsilon} = \langle\nabla f(X), \Delta\rangle + O(\epsilon^2). \tag{18}$$

Multiplying by $\Delta$ and taking expectation yields the result.

The key point in this argument is that the symmetric finite-difference estimator cancels the second-order term in the scalar expansion, leaving only a third-order remainder whose contribution scales as $O(\epsilon^2)$ after division by $2\epsilon$. This is precisely why two-point estimators are often preferred in zeroth-order optimization when accuracy is important. In our setting, this bias control is especially valuable because it shows that constraining perturbations to the tangent space does not distort the local gradient signal; rather, it preserves the same second-order consistency one expects from standard symmetric estimators. The lemma therefore establishes that the RoZO estimator remains asymptotically faithful to the first-order gradient as $\epsilon$ becomes small.

It is also useful to interpret the result geometrically. Since $\Delta$ lies in the tangent space, the estimator only probes the objective along locally feasible low-rank directions. The lemma implies that this restriction does not introduce a first-order bias penalty. In other words, although RoZO searches in a lower-dimensional structured space, it does not sacrifice the local correctness of the gradient approximation. This observation is important for the overall paper narrative: the advantage of RoZO does not come from changing the optimization objective, but from approximating the same objective more efficiently within the right local geometry.

**Lemma 2 (Variance)**   The estimator satisfies

$$\mathbb{E}\|\hat{\nabla}f(X)\|^2 \le O(r(m+n)). \tag{19}$$

*Proof.* The perturbation $\Delta$ lies in the tangent space of the LoRA manifold, which has intrinsic dimension proportional to $r(m+n)$. Since the estimator is linear in $\Delta$, its variance depends on the intrinsic dimension rather than the ambient dimension. Therefore, the variance scales as $O(r(m+n))$.

The main significance of this lemma is that it captures the central theoretical advantage of RoZO. In a naive zeroth-order estimator defined over the full parameter space, the second moment typically scales with the ambient dimension $d$, which is prohibitively large for modern language models. RoZO avoids this unfavorable dependence because the tangent perturbation is parameterized through low-rank factors. Intuitively, the estimator no longer wastes queries exploring directions that violate the low-rank adaptation structure. Instead, it concentrates the search on a subspace whose dimension grows only linearly with $r(m+n)$, which is dramatically smaller than $mn$ in the regimes of interest.

This dimensionality reduction is not merely a counting argument; it has direct optimization consequences. Lower second moment implies lower estimator noise, which in turn stabilizes the descent process and reduces the number of queries required to achieve a given level of progress. This theoretical observation is exactly consistent with the empirical variance curves reported in the main text. From a proof perspective, Lemma 2 is the term that replaces the unfavorable ambient-dimension dependence of standard zeroth-order convergence results. Once inserted into the descent inequality, it leads directly to a sharper optimization rate.

**Theorem 1 (Convergence)**   Let $\{X_t\}$ be generated by RoZO with step size $\alpha$. Then

$$\frac{1}{T}\sum_{t=1}^{T}\mathbb{E}\|\nabla f(X_t)\|^2 \le O\left(\frac{f(X_0)-f^*}{\alpha T}\right) + O(\alpha r(m+n)) + O(\epsilon^2). \tag{20}$$

*Proof.* By smoothness,

$$f(X_{t+1}) \le f(X_t) - \alpha\langle\nabla f(X_t), \hat{\nabla}f(X_t)\rangle + \frac{L\alpha^2}{2}\|\hat{\nabla}f(X_t)\|^2. \tag{21}$$

Taking expectation and applying Lemma 1 and Lemma 2 yields

$$\mathbb{E}[f(X_{t+1})] \le \mathbb{E}[f(X_t)] - \alpha\|\nabla f(X_t)\|^2 + O(\alpha\epsilon^2) + O(\alpha^2 r(m+n)). \tag{22}$$

Summing over $t$ and rearranging yields the result.

The theorem takes the familiar form of a non-convex stochastic convergence bound: the average squared gradient norm is controlled by an optimization term, a variance-related term, and a bias-related term. What distinguishes the present result is the structure of the variance term. Instead of depending on the ambient dimensionality, it depends on the intrinsic low-rank tangent-space dimension. This makes the theorem directly relevant to the large-model setting, where the difference between $d$ and $r(m+n)$ can be enormous. The result therefore formalizes the intuition that RoZO is not just a more stable heuristic, but a method whose sample efficiency is theoretically improved by geometric restriction.

The three terms on the right-hand side also admit a useful interpretation. The first term decreases with the number of iterations and represents optimization progress. The second term reflects stochastic estimation noise and becomes smaller when the step size is chosen appropriately. The third term captures finite-difference bias and vanishes as $\epsilon$ decreases. Thus, the theorem clearly exposes the practical trade-offs faced by the algorithm: smaller $\epsilon$ improves bias, smaller $\alpha$ reduces noise amplification, and larger $T$ improves optimization accuracy. This decomposition aligns naturally with the hyperparameter sensitivity results in the main text, where perturbation radius and query budget were shown to produce precisely these kinds of trade-offs.

Finally, the theorem explains why RoZO can remain competitive even in black-box settings where only function evaluations are available. Standard zeroth-order methods often degrade rapidly with model size

because their variance scales with the ambient dimension. By replacing that dependence with an intrinsic geometric one, RoZO preserves the essential form of stochastic non-convex convergence while making it practically meaningful for low-rank LLM adaptation. This is the core theoretical justification for the method.

### A.5 Geometry of the LoRA Manifold

We now formalize the geometric structure that underlies the proposed algorithm. The purpose of this subsection is to show that the low-rank LoRA parameterization is naturally associated with a smooth manifold and that the operations used by RoZO, such as tangent perturbation and retraction, are mathematically well-motivated. This is important because without an explicit geometric formulation, one might view the algorithm merely as a low-rank heuristic. The analysis below clarifies that the restriction to structured perturbations is grounded in a legitimate manifold optimization perspective.

**Manifold Definition** The LoRA manifold is defined as

$$\mathcal{M}_r = \{BA : B \in \mathbb{R}^{m \times r}, A \in \mathbb{R}^{r \times n}\}. \tag{23}$$

This set forms a smooth manifold embedded in $\mathbb{R}^{m \times n}$.

This definition captures the set of all matrices that can be represented through a rank-$r$ factorization. In the LoRA setting, such matrices correspond exactly to trainable low-rank updates added to a frozen pretrained weight matrix. Thus, optimizing over $\mathcal{M}_r$ is not an artificial mathematical abstraction, but a direct description of the parameter-efficient fine-tuning space. The manifold viewpoint becomes especially useful when we want to reason about local feasible directions, because not every arbitrary perturbation in $\mathbb{R}^{m \times n}$ preserves the intended low-rank structure. By defining $\mathcal{M}_r$ explicitly, we can distinguish between valid low-rank update directions and directions that only exist in the ambient Euclidean space.

Another advantage of this formalization is conceptual clarity. Many previous low-rank zeroth-order methods exploit low-dimensional structure only implicitly by sampling random low-rank perturbations. RoZO instead treats the low-rank parameterization as a geometric object in its own right. This shift in viewpoint enables the use of tangent spaces, transport, and retraction in a principled manner. As a result, the algorithm is best understood not as a random perturbation method with low-rank flavor, but as a genuine Riemannian zeroth-order method tailored to LoRA-style adaptation.

**Tangent Space** At a point $(A, B)$, the tangent space is

$$T_{(A,B)}\mathcal{M}_r = \{B\Delta A + \Delta BA\}. \tag{24}$$

This characterization ensures that updates preserve the low-rank structure locally.

The tangent space describes the first-order feasible directions in which one can move while staying on the manifold up to local approximation. This is exactly the right object for constructing zeroth-order perturbations, because finite-difference estimators are fundamentally local. If perturbations are chosen outside the tangent space, then the algorithm explores directions that are not naturally compatible with the low-rank parameterization, which can increase variance and introduce instability. By contrast, tangent-space perturbations remain aligned with the local geometry of the update space, ensuring that the estimator probes only meaningful low-rank directions.

The form $B\Delta A + \Delta BA$ is also intuitive from the factorized parameterization itself. A small change in $A$ while keeping $B$ fixed contributes the term $B\Delta A$, and a small change in $B$ while keeping $A$ fixed contributes the term $\Delta BA$. Their sum therefore captures all first-order variations induced by perturbing the factors. This representation makes clear why the intrinsic degrees of freedom scale with $r(m + n)$ rather than $mn$. That same dimensionality count reappears in the variance bound of the previous subsection, linking the geometry directly to the optimization analysis.

**Retraction** To ensure that updates remain on the manifold, we use a retraction operator defined as

$$\text{Retract}(X + \Delta) = \arg \min_{Y \in \mathcal{M}_r} \|Y - (X + \Delta)\|. \tag{25}$$

Retraction is needed because even if the update direction lies in the tangent space, a finite step along that direction does not necessarily land exactly on the manifold. This is a standard issue in manifold optimization. The tangent space gives only a local linear approximation of the manifold, so after taking a step one must map the result back to the feasible set. In RoZO, retraction plays precisely this role: it converts an off-manifold intermediate point into a nearby low-rank point that remains consistent with the LoRA parameterization. Without this operation, the iterates could gradually drift away from the intended rank-constrained structure, undermining both efficiency and theoretical validity.

From a practical standpoint, retraction is also preferable to exact geodesic updates because it is computationally much simpler while still preserving the local geometry to second order. This makes it a natural choice for large-scale fine-tuning. The theoretical importance of retraction is that it allows one to combine local tangent-space reasoning with globally feasible iterates. In other words, tangent perturbation governs how the search direction is chosen, while retraction guarantees that the resulting parameters remain within the model class being optimized.

**Lemma 3 (Retraction Error)**    The retraction satisfies

$$\text{Retract}(X + \Delta) = X + \Delta + O(\|\Delta\|^2). \tag{26}$$

This implies that retraction is a second-order approximation to the exponential map.

This lemma is important because it shows that retraction does not distort the update at first order. If the retraction error were of order $O(\|\Delta\|)$, then the update direction itself could be significantly altered, invalidating the local gradient approximation used in the convergence proof. The second-order error guarantee ensures that for sufficiently small steps, the retracted point behaves like the ideal manifold-consistent move up to higher-order terms. Consequently, one can safely carry out the optimization analysis using tangent-space arguments without introducing a dominant additional error term.

The lemma also clarifies the conceptual role of retraction in RoZO. Its purpose is not to redefine the descent direction, but to enforce feasibility while preserving local optimization information. This means that the advantages of RoZO still come from tangent-space restriction and geometry-aware perturbation; retraction simply ensures that those advantages are realized over many iterations without violating the low-rank structure. In this sense, retraction is a compatibility mechanism between local linearization and global manifold feasibility.

## A.6  Properties of Parallel Transport

We next analyze the role of parallel transport in maintaining consistency of momentum across iterations. This is a key ingredient of RoZO-M and is one of the main differences between our approach and Euclidean momentum schemes applied to low-rank parameterizations. Because tangent spaces at different points on the manifold are not identical, a momentum vector computed at one iterate cannot be naively reused at the next iterate without potentially introducing geometric inconsistency. Parallel transport resolves this issue by mapping vectors from one tangent space to another in a way that approximately preserves their geometric meaning.

**Lemma 4 (Norm Preservation)**    Parallel transport approximately preserves vector norms:

$$\|\text{Transport}(v)\| = \|v\| + O(\|v\| \cdot \|\Delta\|). \tag{27}$$

This property shows that transport does not substantially inflate or shrink the magnitude of the momentum vector when the step between iterates is small. Such control is important for optimization stability. If transported momentum vectors systematically changed norm, then the effective learning dynamics could become erratic even when the underlying stochastic estimator is well-behaved. The lemma therefore justifies using transported momentum in place of naive Euclidean accumulation. In local terms, the transported vector remains close in magnitude to the original one, so the optimizer retains a consistent notion of step size as it moves across the manifold.

From an intuitive standpoint, norm preservation means that the algorithm carries magnitude information from one iteration to the next without significant distortion. This complements the convergence analysis in Appendix A.4, where stability of updates is essential for obtaining meaningful descent. In the context of large language model fine-tuning, where optimization trajectories can be long and noisy, maintaining such consistency is especially important.

**Lemma 5 (Angle Preservation)** Parallel transport approximately preserves inner products:

$$\langle \text{Transport}(u), \text{Transport}(v) \rangle = \langle u, v \rangle + O(\|\Delta\|). \tag{28}$$

Angle preservation is arguably even more important than norm preservation, because momentum is fundamentally a directional mechanism. Its purpose is to accumulate compatible descent directions over time so that optimization progresses more smoothly and avoids short-term oscillations. If transport were to distort angles substantially, then the accumulated momentum could quickly lose its intended directional meaning. The lemma shows that this does not happen to first order: transported vectors preserve their relative orientation up to a small error proportional to the step size.

This result provides a rigorous explanation for why RoZO-M is more stable than geometry-agnostic momentum variants. In Euclidean schemes applied directly to changing low-rank parameterizations, there is no guarantee that a past momentum vector remains compatible with the current tangent space. Parallel transport addresses exactly this mismatch. As a result, the accumulated momentum in RoZO-M can still be interpreted as a coherent summary of past descent information, rather than as an arbitrary combination of vectors defined in incompatible local spaces.

**Implication** These properties ensure that momentum directions remain stable when moving across tangent spaces. Without transport, naive momentum accumulation would lead to inconsistent updates due to changes in local geometry.

The implication is both theoretical and practical. Theoretically, transport allows us to extend the intuition of momentum-based acceleration from Euclidean optimization to the manifold setting without losing local consistency. Practically, it reduces oscillation and improves late-stage convergence, which is exactly the phenomenon observed in the experimental section when comparing RoZO-M with previous low-rank zeroth-order baselines. In other words, the transport mechanism is not an ornamental geometric addition; it is necessary if one wants momentum to behave as intended under changing tangent spaces.

It is also worth emphasizing that this consistency is achieved without introducing a large computational burden. Since RoZO already operates in the intrinsic low-rank space, transport is applied to low-dimensional structured objects rather than full dense parameter tensors. This preserves the efficiency advantages of the method while enabling a more faithful manifold-aware extension of momentum.

**Discussion** The combination of tangent-space perturbation, retraction, and parallel transport forms a coherent geometric optimization framework. Each component plays a specific role: tangent restriction reduces variance, retraction maintains feasibility, and transport preserves directional information. Together, they enable RoZO to achieve both stability and efficiency in zeroth-order fine-tuning.

Viewed as a whole, these components explain why RoZO should be expected to outperform geometry-agnostic baselines even before one inspects the empirical results. Tangent-space restriction improves the quality of the zeroth-order estimator by reducing the effective search dimension. Retraction ensures that local updates accumulate into a valid global optimization trajectory on the low-rank manifold. Parallel transport allows momentum to remain meaningful across iterations despite changes in local coordinates. None of these mechanisms alone fully explains the behavior of the method; rather, their interaction is what produces a stable, scalable, and theoretically grounded algorithm.

This integrated perspective is important for positioning the contribution of the paper. RoZO is not simply a collection of useful tricks layered on top of a zeroth-order optimizer. It is a unified geometric framework in which each operation is motivated by a specific structural property of low-rank adaptation. The convergence

proof, variance bound, and transport lemmas together show that the method is theoretically coherent, while the experiments demonstrate that this coherence translates into practical gains.

### A.7 Mechanism Analysis

To further understand the underlying mechanism of RoZO, we analyze its behavior from a geometric and spectral perspective. The central hypothesis of this work is that low-rank adaptation naturally lies on a structured manifold, and that respecting this geometry leads to better optimization. While previous sections have demonstrated improvements in variance and performance, they do not directly reveal whether RoZO indeed captures meaningful structural information. To address this, we examine the singular value distribution of updates generated during training.

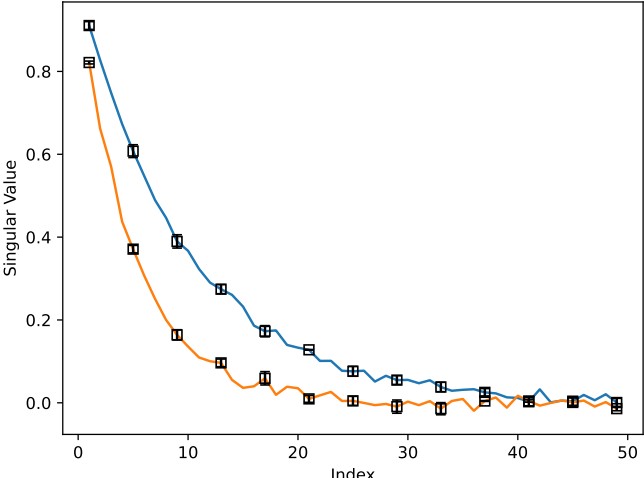

Figure 10: Singular value comparison between RoZO and baseline methods. RoZO maintains a smoother decay profile.

Figure 10 shows that RoZO produces a smoother singular value decay compared to baseline methods. This indicates that update energy is distributed more evenly across multiple directions, rather than collapsing into a few dominant components. Such behavior is desirable in low-rank adaptation, as it reflects a richer and more stable representation of task-specific information. In contrast, the baseline exhibits a sharper decay, suggesting that it fails to preserve meaningful structure during optimization.

The spectral analysis also provides insight into the variance and alignment improvements observed earlier. A smoother singular value distribution implies that perturbations are better aligned with the intrinsic geometry of the parameter space. This reduces the likelihood of exploring irrelevant directions, thereby lowering variance. At the same time, it increases the probability that updates align with true descent directions, leading to improved gradient alignment. Thus, the spectral behavior serves as a unifying explanation for multiple empirical observations.

Overall, the mechanism analysis confirms that the effectiveness of RoZO is rooted in its geometric formulation. By operating within the tangent space of a low-rank manifold, the method reshapes the optimization landscape in a way that favors stable and meaningful updates. This structural advantage distinguishes RoZO from existing zeroth-order methods, which rely on random perturbations and therefore fail to capture the underlying geometry of low-rank adaptation.

