# OpenReview forum: "RoZO: Geometry-Aware Zeroth-Order Fine-Tuning on the Low-Rank Adapters for Black-Box Large Language Models"
_TMLR — Under review for TMLR_

### Review · Reviewer_gPqx · 2026-05-28

**Summary Of Contributions:**

The paper proposes RoZO, a geometry-aware zeroth-order optimization framework for parameter-efficient fine-tuning of large language models. The main idea is to constrain zeroth-order (ZO) perturbations to the tangent space of the low-rank LoRA manifold. The paper further integrates retraction, parallel transport, adaptive preconditioning and trust-region control. The paper claims improved stability, lower estimator variance, and stronger empirical performance than MeZO and LOZO across several NLP benchmarks and model sizes.

Strength:

1.	The topic is important and timely: memory-efficient for fine-tuning LLMs.

2.	The proposed solution of zero-order low-rank LoRA adapters is interesting and may be potential.

3.	The overall empirical performance shows potential.

Weaknesses:

1.	The details for Section 4.1 are missing, even when accounting for the Appendix. First, it is not clear how to compute the tangent space in practice. Second, it seems that the manifold formulation is only geometry-aware w.r.t the dimensionality of the problem, not the objective landscape itself. For example, given 2 problems with the same values of m, n, r, based on Eq.5, they will have identical manifold, hence identical perturbations.

2.	Some experiment details are not clear. See the next section for clarification.

3.	Many other details are unclear. See the Requested Changes section.

**Audience:**

Yes

**Audience Explanation:**

The topic is relevant because zeroth-order optimization, parameter-efficient fine-tuning, and memory-efficient adaptation of LLMs are active areas of interest. The proposed direction - using low-rank geometry to improve ZO fine-tuning, is conceptually interesting and could inspire further work.

**Claims And Evidence:**

No

**Claims Explanation:**

While the reported results are promising, many details are missing, making the results less convincing.

1.	It is not clear whether the hyperparameter setting for RoZO is similar across all experiments. This also raises concerns about reproducibility.

2.	It is not clear why memory experiments (Section 5.2) have to use a different set of problems, while other sections (5.1, 5.3, 5.4) already include a lot of experiments.

3.	On scalability problems (Sections 5.3 and 5.3), it is not clear why there is no RoZO-M variant.

**Requested Changes:**

1.	Can the authors give more details on the formulation of manifold, especially in practice.

2.	Clarify the differences between RoZO and RoZO-M variants. Currently the authors only mention RoZO-M as a momentum variant, which is vague. For example, does RoZO-M include Preconditioning and Trust Region?

3.	Clarify the variants in ablation study, specifically what their alternative components are. For example, in “W/o Parallel Transport”, how does Eq. (9) become? Also, do all variants use the LoRA manifold in Eq. (5)?

4.	Can the authors clarify whether the hyperparameter setting for RoZO is similar across all experiments. If not, can the authors provide a practical guidance on choosing the hyperparameter for new problem?

5.	I think the authors should add the memory usage (similar to Table 1) for all experiments that were used in Sections 5.1, 5.3, 5.4? Also, can the authors add RoZO-M for scalability experiments (Sections 5.3, 5.4)?

---

> ### Author Response · Authors · 2026-07-01
>
> We sincerely appreciate the reviewers’ careful evaluation and constructive suggestions. In response to the issues they identified, we have revised the manuscript and uploaded the updated PDF to the TMLR submission system. For ease of review, all newly added, materially revised, or substantially rewritten text is highlighted in blue in the revised version. We provide below a point-by-point response to the reviewers’ comments and summarize the corresponding changes in the manuscript.
>
> **Manifold formulation and practical computation.** We added implementation details in Sec. 4.1. RoZO samples factor perturbations $(\Delta A_t, \Delta B_t)$, constructs the tangent direction $\Delta_t = B^t \Delta A_t + \Delta B_t A^t$, and evaluates $F(W_0 + X^t \pm \epsilon \Delta_t; \xi^t)$. This avoids materializing dense perturbation matrices. We also added the standard tangent-projection formula for cases in which a dense candidate direction must be projected.
>
> **Does the geometry encode the objective landscape?** We clarified that the low-rank manifold does not by itself encode the full task objective. Two problems with the same $(m,n,r)$ share the same feasible manifold, but their trajectories differ through the current LoRA factors, finite-difference loss values, and trust-region decisions. Thus, RoZO is geometry-aware with respect to feasible low-rank update directions, while the objective still determines which directions are useful.
>
> **RoZO vs. RoZO-M.** We clarified in Secs. 4.3 and 5 that RoZO-M is RoZO augmented with transported momentum. Unless an ablation explicitly removes a component, RoZO-M retains the same tangent estimator, retraction, adaptive preconditioning, and trust-region control as RoZO.
>
> **Ablation variants.** We revised Sec. 7 to specify precisely what each ablation changes. In particular, the “w/o Parallel Transport” variant replaces transported momentum with Euclidean accumulation,
> $$
> m^t = \gamma m^{t-1} + (1-\gamma)\widehat{\operatorname{grad}} f(X^t),
> $$
> followed by the same retraction. We also clarified that all ablation variants retain the LoRA manifold and tangent estimator unless explicitly stated otherwise.
>
> **Hyperparameter reproducibility.** We added a practical hyperparameter guide in Sec. 6, including shared candidate grids, validation-only selection, default values for $r$, $q$, $\epsilon$, $\gamma$, and $\beta_2$, as well as guidance for larger models and unstable estimates. We also clarified that baselines are tuned under comparable query and FLOP budgets.
>
> **Memory reporting and RoZO-M scalability.** We added the memory-measurement protocol and moved the memory table to Sec. 5.2. We also clarified that the larger OPT-model tables currently report base RoZO to isolate the tangent-space estimator under matched query budgets. RoZO-M differs only through its tangent-space momentum state and therefore has the same asymptotic memory order. We agree that full per-task memory logs and RoZO-M results for the larger-model scalability experiments would further strengthen the paper, and we will include these results once the corresponding large-model runs are complete.

---

### Review · Reviewer_YXBE · 2026-06-01

**Summary Of Contributions:**

This work proposes a fine-tuning approach for low-rank adapters in language model without relying on gradient estimates by back propagation. The prior work on adapter tuning demands back-propagation for estimating the updates to adapters, but zeroth-order optimization removes the constraints by relying on forward differentiation. However, still issues remain for zeroth-order optimization in that it still demands memory for storing perturbation matrices and instability in training. This work alleviates the issue by focusing on optimization in Riemann space replying on the LoRA adapter decomposition, preserving momentum in the space and adapting the perturbation and learning rate hyperparameters.

Strengths:
- An interesting approach for zeroth-order optimization for LoRA adapter.
- Experimental results seem to be solid.

Weaknesses:
- The manuscript is very hard to follow mainly because of the paper structure. It is apparent that this work needs substantial rewrite to improve the readability.

**Audience:**

No

**Audience Explanation:**

This work needs further rewrite for the submisison quality.

**Claims And Evidence:**

No

**Claims Explanation:**

Given that the manuscript is very hard to follow, it is not clear whether my understanding is correct. No explanation exists for the math notations and it is not clear whether the proposed approach is well articulated. Tables are placed at random places, causing issues of the readability and no convergence analysis presented in section 4.5.

**Requested Changes:**

- Explain the math notations in section 3 and 4. In particular, Equation 2 through 4 are the most important to establish the methodology, though, the notations are not well described. They are even not aligned with prior studies mentioned in section 3.1, regarding CGE and RGE. Similarly Equation 7 and 8 are not well described, and thus, the core idea is not articulated well.
- Restructure the manuscript and place tables at appropriate places.
- No convergence analysis exists in section 4.5, although promised to discuss.

---

> ### Author Response · Authors · 2026-07-01
>
> We sincerely thank the reviewers for their careful reading and constructive feedback. We have revised the manuscript in response to the issues raised and uploaded the revised PDF to the TMLR submission page. To facilitate review, all newly added, substantially revised, or rewritten material is highlighted in blue in the revised manuscript. Below, we respond point by point to the reviewers' comments and summarize the corresponding revisions.
>
> **Notation in Sectinotation around Eqs. 2--4 and Eqs. 7--8 was underspecified.** We have revised Sec. 3.1 to explicitly define $D_\epsilon F(X;\xi;u)$, $X$, $\mathbf{X}$, $\xi$, $e_i$, $z$, $\epsilon$, and $q$, and to write CGE and RGE in their standard forms. We also revised Sec. 4.1--4.2 to define $W_0$, $X=BA$, $\mathcal{M}_r$, $T_X\mathcal{M}_r$, $\Delta_t$, and the scalar two-point directional derivative used by RoZO.
>
> **Paper structure and table placement.** We moved the NLI figure, memory table, OPT-13B table, larger-model table, and ablation table into the experimental subsections where they are discussed. This should remove the previous impression that tables appear randomly inside the method section.
>
> **Convergence analysis in Sec. 4.5.** We expanded Sec. 4.5 from a high-level claim into a concrete statement of assumptions, estimator bias, variance scaling, and the resulting stationarity bound. The full proof remains in Appendix A.4, but the main text now states the rate and explains why the dependence changes from the ambient dimension $d$ to the intrinsic LoRA tangent dimension $d_r=r(m+n-r)$.